# Compendium of Methods to Uncover RNA-Protein Interactions In Vivo

**DOI:** 10.3390/mps4010022

**Published:** 2021-03-19

**Authors:** Mrinmoyee Majumder, Viswanathan Palanisamy

**Affiliations:** Department of Biochemistry and Molecular Biology, College of Medicine, Medical University of South Carolina, Charleston, SC 29425, USA; visu@musc.edu

**Keywords:** RNA, ribonucleoproteins, RNA–protein interactions, gene expression and post-transcriptional gene regulation

## Abstract

Control of gene expression is critical in shaping the pro-and eukaryotic organisms’ genotype and phenotype. The gene expression regulatory pathways solely rely on protein–protein and protein–nucleic acid interactions, which determine the fate of the nucleic acids. RNA–protein interactions play a significant role in co- and post-transcriptional regulation to control gene expression. RNA-binding proteins (RBPs) are a diverse group of macromolecules that bind to RNA and play an essential role in RNA biology by regulating pre-mRNA processing, maturation, nuclear transport, stability, and translation. Hence, the studies aimed at investigating RNA–protein interactions are essential to advance our knowledge in gene expression patterns associated with health and disease. Here we discuss the long-established and current technologies that are widely used to study RNA–protein interactions in vivo. We also present the advantages and disadvantages of each method discussed in the review.

## 1. Introduction

The majority of the nascent RNA transcripts are associated with and regulated by several proteins. Almost all RNAs, irrespective of their structure and function, interact with one or more protein partners to function correctly [1,2]. Both RNA and protein can undergo structural changes, permitting them to form stable complexes during RNA processing, and these conformational changes are commonly called induced fits [3]. There are three induced fit models or conformational changes that can occur in RNA, protein, or both, which facilitate the achievement of proper biological functions such as: (I) protein-induced RNA folding, (II) RNA-induced protein folding, and (III) co-induced folding (Figure 1). The basic stem-loop structure of RNA forms diverse structures with A-U and G-C Watson–Crick pairs and G-U wobble base pairs [4]. These RNA structures facilitate the association with multiple RNA-binding proteins and form complexes, subsequently carrying out their biological functions. The identification and characterization of the RNA-protein complexes are performed with numerous in vivo and in vitro methodologies. Although many reported in vitro methods analyze RNA–protein interactions, the current compilation of in vivo approaches is minimal. In reviewing in vivo methodologies to study RNA-protein interactions, we highlight each method’s advantages and disadvantages to identify and characterize RNA-protein complexes. 

Genome-wide association studies of RNA–protein interactions

Recent advances in molecular biology techniques have expanded into genome-wide studies as “OMICS” dealing with individual genetic components such as genomics for DNA and transcriptomics for RNA. These technologies are often referred to as high-throughput sequencing (HTS) used to screen multiple targets and samples in a given time. Interestingly, when combined with biochemistry and molecular biology assays, HTS provides detailed knowledge of the target genes and delineates the model organism’s biological functions through a network or pathway analysis [5]. Current methodologies used to identify the components of RNA-protein complexes include RNA immunoprecipitation (RIP) or UV/chemical crosslinking and IP (CLIP), followed by either microarray or HTS profiling of the bound RNAs [6,7,8]. Another recent technique, in vivo photoactivatable-ribonucleoside-enhanced crosslinking and immunoprecipitation (iPAR-CLIP), expands crosslinking photoreactive RNA molecules followed by HTS analysis. Two modifications of CLIP have been reported, individual-nucleotide resolution cross-linking and immunoprecipitation (iCLIP), which uses the reverse transcriptase method to map individual nucleotide-protein interactions, and m6A individual-nucleotide-resolution cross-linking and immunoprecipitation (miCLIP), which modifies an RNA methylase, thereby allowing its binding sites in living cells to be mapped [9,10]. The following section describes detailed analyses using RIP and the different CLIP methods for genome-wide screening of RNA–protein interactions.

### 1.1. RNA Immunoprecipitation Method to Study Ribonucleoprotein and RNA Molecule

Immunoprecipitation of proteins that form complex with RNA is a commonly used method to detect the unknown RNA components. Based on protein-RNA interaction, separation of protein-RNA complex in a natural condition or under a chemical or UV crosslinking approach requires a strong association between RNA and protein at the intracellular level. The proteins can be subjected to IP using specific antibodies against them, or affinity tagged antibodies against the particular protein can be used to isolate the entire RNA-protein complex. Identification of the RNA molecules is the primary purpose of this strategy, where the untagged proteins or control Immunoglobulin-G (IgG) usually serve as an experimental control for this strategy. 

*Affinity tags used for protein pull-down:* Numerous affinity tags, such as c-myc, calmodulin-binding peptide CBP, FLAG, Histidine (His), Glutathione-S Transferase (GST), and the Tandem Affinity Purification (TAP) [11,12,13], are available for tagging and tracing of proteins both in vitro and in vivo. The RNA tags used more efficiently are Strepto, Streptavidin, and Sephadex [14,15,16]. TAP protein purification was first discovered by the Seraphin group [12,17]. It was based on a fusion cassette containing calmodulin-binding peptide CBP, a TEV cleavage site, and two IgG binding sites of *Staphylococcus aureus* protein A, cloned at either the 5’ or 3’ end of the desired gene in the chromosome. Incorporating the TAP tag in the protein does not alter its expression level and is therefore useful for the purification and characterization of protein-RNA complexes. A modified TAP protocol includes a TEV protease cleavage site replacement with a 3C protease, and CBP is substituted with 6X His [18,19]. Histidine is also a commonly used tag for protein purification and additional binding assays to detect protein–protein and protein–RNA interactions. A polyhistidine-tag is an amino acid sequence in proteins that contain at least five or six histidine (His) residues and can be placed at the N- or C-terminus of the protein. Endopeptidases (like Thrombin) can remove the His-tag when present after the tag. Alternatively, an exopeptidase can be applied depending on His tag’s proximity. The His-tagged protein and associated components can be purified by a passage through Ni-NTA resin, increasing imidazole concentration [20,21,22]. 

*Tagging RNA molecules:* RNA affinity tags (aptamers) like Strepto, Streptavidin S1, and Sephadex D8 are commonly and successfully used to isolate the in vivo RNA-protein complexes. However, due to an added tag to the macromolecules, the RNA native fold distortions may occur. In silico RNA secondary structure must be verified after adding a tag to the RNA by running RNA probing experiments [23,24]. The Strepto tag contains an RNA sequence 46 nucleotide binds to the antibiotics streptomycin [14], used with yeast group II introns and viral RNAs to identify novel proteins that bind to a set of RNAs in yeast and viruses. The hybrid RNA can be immobilized on a streptomycin affinity column, and the proteins will be selected from cellular or nuclear extracts [25]. The Streptavidin S1 consists of 44 nucleotides, and Sephadex D8 contains 33 nucleotides of RNA sequences, specifically bind to the streptavidin protein and dextran, respectively [15,16]. Both RNA motifs are incorporated into the RNA molecule of RNase P and select RNP complexes with either the streptavidin-agarose or Sephadex G-200 matrix. Sephadex G-200 is inexpensive, and the concentration of ligand on the beads is almost infinite, which makes this purification method very useful when large starting quantities are employed. However, unlike streptavidin-agarose, the affinity of RNA for the ligand is not high, which increases the chance of loss of the bound complex to the G-200 resin after extensive washing.

Although the above-mentioned RNA-aptamer approaches are relatively adaptable and signify an efficient tool for detecting the RNA-protein complexes, a significant disadvantage is the indirect nature of the interaction. These interactions may result from the tagged RNA’s expression in trans that cross-reacts with native RNAs and RNPs. A few other techniques besides aptamers have also been used to determine the in vivo RNP complexes. Antisense oligonucleotide (ASO) probes like biotinylated 2-*O*-methylated RNA oligos have been applied to RNAs for the direct purification of abundantly-expressed RNP complexes such as the spliceosome, telomerase, and long ncRNAs (lncRNAs) from human cells [26,27]. Additionally, a two-step procedure, the tandem RNA isolation procedure (TRIP), is based on DNA/RNA ASOs, and it recovers native mRNA-protein complexes from cells with high selectivity [26]. Primarily, this method was tested on specific mRNAs present in yeast, nematodes, and human cells. Enhancing the poly(A) RNAs from a cell lysate using oligo(dT)_25_ beads following a purification step to capture the particular mRNAs from poly(A) RNAs using 3′-biotinylated 2-*O*-methylated RNA ASOs are two necessary steps of the TRIP method. The isolated RNAs and proteins can be readily identified by RT-PCR or protein blots; however, global identification of these macromolecules can also be done by RNA sequencing and mass spectrometry, respectively.

*Isolation of cellular RNA-protein complexes:* The plasmids carrying the tagged protein or RNA are introduced into the cells for pull-down, for which empty vectors are used as a control. Pull-down experiments are performed after the cell lysis, after which lysate is passed through an affinity column resin or matrix containing an antibody against the tag or protein or RNA of interest. Finally, the native complex can be eluted using specific chelating agents, like imidazole, mostly used for His-tagged proteins or streptavidin for biotin tags on RNA. After purification, the components of the complex are characterized by liquid chromatography-assisted mass spectrometry (LC/MS). The LC/MS is commonly used to identify proteins; still, with advanced technology, MS offers several advantages for RNA analysis, evolving from its ability to provide mass and sequence information from limited amounts of the sample [28,29]. Once the sequence is identified, the unknown protein bound to the tagged RNA will be further characterized by in vivo and/or in vitro approaches to determine the specific motifs that attach to the RNA, affinity of binding, function of the RNP, and nucleotide modifications. Alternatively, the RNA can be extracted for northern blot analysis, primer extension, and further cDNA synthesis for cloning and sequencing [30].

### 1.2. Crosslinking to Map RNA–Protein Interactions by Using Immunoprecipitation CLIP

In the study of RNA-protein interaction in vivo, it is critical to use methods that rapidly halt the RNA-protein complexes and prevent their rearrangement during cell lysis. Multiple crosslinking agents are used, but reversible crosslinking agents are the most useful because they facilitate the subsequent characterization of the interacting partner molecules. Cross-linking can stabilize RNA and protein complexes and can be a valuable tool for analyzing sequence-specific interactions. This can be achieved using UV light or chemical crosslinkers [3,31,32]. Some commonly used chemical crosslinkers are dimethyl suberimidate, the imidoester crosslinker BS3, a NHS-ester crosslinker, and formaldehyde. For in vivo crosslinking of protein-RNA complexes using photoreactive amino acid analogs, the cultured cells are grown with photoreactive diazirine analogs that react with leucine and methionine and are incorporated into proteins. After treatment with UV light, the diazirines are activated and bound to interacting proteins that are within a few angstroms of the photoreactive amino acid analog [33].

#### Photo Cross-Linking/U.V. Crosslinking

In vivo UV crosslinking has the benefit of analyzing RNA–protein interactions in their native states. However, it requires prolonged exposure, thus permitting the redistribution of the components and crosslinking of the UV-damaged molecules. The in vivo UV crosslinking technique is based on UV light’s ability to crosslink proteins to RNA in living cells covalently. Photoreactive crosslinking is commonly used in cell lines that express tagged proteins. However, the cells containing endogenous RNA-protein complexes, for which an antibody against the protein is required, can be analyzed by this method. In proteins, the amino acids cysteine, tyrosine, phenylalanine, arginine, lysine, and tryptophan most readily crosslink to poly-U.

Additionally, it is often noticed that pyrimidines have higher efficiency to be crosslinked than purines and RNA, with uridine residues compared to thymidine in DNA, shows greater reactivity when comparing addition to cysteine [34,35]. Besides crosslinking efficiency, antibody specificity against the interacting protein is a critical factor in these assays. This can be overcome by adding epitope tags to the interacting proteins; however, it may restrict the proper folding of the protein inside the cells or affect the protein’s ability to interact with specific RNAs.

The UV-irradiated cells are harvested with lysis buffer containing sodium dodecyl sulfate (SDS) to liberate genomic DNA and subsequent heating and passing through a spin column to remove the DNA, which in turn increases the RNA yield. This step can be substituted with sonication, which may lead to significant RNA degradation. Next, the lysate is cleared by centrifugation for further analysis with a specific antibody bound to beads (agarose or magnetic) to detect the proteins [36,37,38,39,40]. Before western blotting, the proteins are stripped from the beads by boiling in standard SDS loading buffer or eluted with a buffer with low pH. The input, supernatant, and beads are treated with proteinase K, then with phenol:chloroform: isoamyl alcohol mixture, and precipitated with ice-cold ethanol to obtain the RNA. Northern blots or RNase protection assays can be performed to analyze the RNA. Control RNA is used as a loading standard to observe the recovery.

Advantages:This procedure does not require any chemical agent, thus avoiding disturbance of the RNA-protein interface.UV irradiation is a zero-length crosslinker, which creates the RNA-protein crosslinks at their point of contact, thus displaying a direct RNA-protein interaction.

Disadvantages:RNA-protein interfaces must be compatible with the chemistry of the photocrosslinking reaction. In addition to potential amino acid biases, there is a preference for uridine to act as a crosslinking base [41].The efficiency of these crosslinks is often less than that of chemical crosslinking.The covalent bonds created by crosslinking are irreversible; therefore, small adducts persist after proteinase K removal of the bound proteins, which inhibit the reverse transcriptase elongation process. Thus, RT-PCR may not be performed to analyze the RNA.

### 1.3. Chemical Crosslinking

#### Formaldehyde Crosslinking Followed by RNA IP (RIP)

Formaldehyde crosslinking is widely used for chromatin immunoprecipitation ChIP [31] and RNP immuno-precipitation RIP assays. Formaldehyde is a reversible crosslinking agent. Generally, crosslinks have specific contact points, forming residue-residue contacts and increasing the strength of these contacts. These crosslinks usually are very stable but can sometimes lose their stability at elevated temperatures. The most common technique used to study RNA–protein interactions in vivo is a formaldehyde crosslinking RNA immuno-precipitation RIP strategy developed by Garcia-Blanco and colleagues [42]. 

Based on chromatin immunoprecipitation ChIP protocols [42,43], chemical crosslinking with formaldehyde has been established to capture complexes formed between RNA and protein in vivo. The success of the RIP assay lies in the crosslinking efficiency. Excessive crosslinking may result in a loss of material due to low solubilization or masking of the epitopes recognized during the IP. However, suboptimal crosslinking may lead to incomplete fixation, thus resulting in fewer crosslinked complexes. The crosslinked proteins are immunoprecipitated with antibodies specific to the protein or the tag attached to the protein. After washing, the RNA, eluted from beads, may be characterized by northern blotting or RNase protection assays [44]. Simultaneously, after the proteins are immunoprecipitated, the crosslinks are reversed by prolonged heating of the samples at a high temperature, and RNA is purified using standard procedures. RNA is analyzed by qPCR after the cDNAs are generated by reverse transcription (RT).

Advantages:The crosslinks are reversible. Therefore, specificity can be achieved by detecting the bound RNA by RT-PCR.This assay preferentially works for nucleophilic lysine residues, which are readily crosslinked by formaldehyde [45].

Disadvantages:The failure to detect the RT-PCR signal can be due to incomplete reversal of the crosslinks or crosslinking involving the antibody’s epitope, thus making it inaccessible to the antibody used during the IP.Additionally, besides protein–RNA interactions, protein–protein interactions can also result from formaldehyde crosslinking, making the detection of the direct RNA-protein complexes from the protein–protein RNA complexes difficult.

### 1.4. CLIP Methods Used to Identify RNA-Protein Interaction Partners

#### 1.4.1. HITS-CLIP (High-Throughput Sequencing-CLIP)

HITS-CLIP (Figure 2) can be applied to protein-RNA crosslinks that are generated at a wavelength of 254nm. Once the cells are lysed, the RNA is partially digested with RNase and then subjected to IP using the known protein. Next, alkaline phosphatase is used for 3’ dephosphorylation to ligate a radiolabeled linker at the 3’ end. A 3’ linker is added for cases in which direct 5’ adapter ligation with polynucleotide kinase PNK produces a high background signal. After the linker is ligated, 5’ phosphorylation is performed by PNK. Next, the complexes will be eluted from the beads and separated on SDS-PAGE, followed by the retrieval of the RNA, RT-PCR, and high-throughput sequencing of the library. This method was developed in Darnell’s laboratory to identify genome-wide binding sites for the Nova protein neuron-specific splicing factor [2]. HITS-CLIP has provided genome-wide confirmation of the bioinformatically predicted Nova-regulated splicing map. It has confirmed the hypothesis that Nova binding to its targets ultimately determines the fate of splicing regulation experimentally identifying the genome-wide Nova binding sites. Interestingly, the same study has also shown that this technique determined Nova binding to alternative polyadenylation sites in the 3’ UTR of its target, thus regulating the outcome of polyadenylation.

Advantages:The method produces a large number of high-throughput sequencing reads for analysis.Thymidine deletion or mutation during RT of the crosslinked site can help determine the nucleotides with high accuracy, and adapter adenylation is not needed for this method.

Disadvantages:UV cross-linking efficiency may be low.A large number of reads can often produce high noise levels, resulting in difficulty in data interpretation.The on-bead ligation can be less efficient for small RNAs, including miRNAs.The reverse transcriptase may stall at the crosslinked site, resulting in a truncated cDNA that does not contain the 5’-adapter and ultimately fails to produce a PCR product.

#### 1.4.2. PAR-CLIP 

Although both photoactivatable ribonucleoside-enhanced CLIP (PAR-CLIP) (Figure 2) and HITS-CLIP approaches use UV light to crosslink proteins to RNA, the significant difference between the two techniques lies in the crosslinking process. Unlike HITS-CLIP, the 4-thio-uridine 4-SU and 6-thioguanosine 6-SG integrated into the RNA during transcription after the cells were exposed to UV at 365nm. A significant advantage of this method is that, unlike regular CLIP, it can precisely identify the binding sites on RNA with high specificity, owing to the RNA labeling method, which generates mutations T to C conversion for 4-SU and G to A for 6-SG at the crosslinked sites during cDNA formation [46].

Advantages:T to C or G to A conversion during RT of the crosslinked site can help determine the nucleotides with high accuracy and separate the labeled RNAs from the unlabeled RNAs.

Disadvantages:4-SU analogs are expensive. Thus, this procedure is not cost-effective.The 4-SU analogs are not always tolerated by the cells.UV cross-linking may be insufficient.

#### 1.4.3. iCLIP

Similar to HITS-CLIP, iCLIP (Figure 2), or individual-nucleotide resolution, CLIP utilizes crosslinking at 254 nm, followed by IP, 3’ adapter ligation discussed above under HITS-CLIP and high-throughput sequencing [9]. However, instead of 5’ adapter ligation, iCLIP uses RT followed by circularization of the cDNA and is more potent to handle a truncated cDNA. This method is regularly used to analyze alternative splicing and polyadenylation signals at the target sites [9].

Advantages:The method is viable under most experimental conditions and often in cases where PAR-CLIP is not feasible, e.g., animal tissues.HITS-CLIP, thymidine deletion, or mutation during the crosslinking of the sites can help determine the nucleotides with high accuracy.Circularization of the cDNA, instead of 5’ adapter ligation, might prove to be more efficient.

Disadvantages:This method does not have many disadvantages. However, the protocol is exceptionally specialized and long, requiring several enzymatic and precipitation steps. Also, the circularization step of iCLIP can be an unreliable reaction at times.

#### 1.4.4. eCLIP

The eCLIP (Figure 2) maps the binding sites of RBPs on their target RNAs using a modified iCLIP protocol, refining efficacy and decreasing execution complexity [47]. This method’s trademark is the ligation of barcoded single-stranded DNA adapters, which significantly reduces the amplification bias. Additionally, UV cross-linking may be low, and interactions near the 3′ end of an RNA may be challenging to identify because reverse transcriptase stops at the cross-linked nucleotide. First, the protein of interest and RNA are UV-crosslinked, followed by cell lysis and RNase I digestion. Next, the RNP complexes are immunoprecipitated and ligated to an RNA adapter on the 3’ end of the target RNA. The bound protein is removed by proteinase K digestion. The RNA is reverse-transcribed, and the resulting cDNA is ligated to single-stranded DNA adapters on the 3’ end containing either an N5 or N10 sequence to serve as unique identifiers against PCR duplicates. In the end, the paired-end cDNA fragments are amplified and sequenced.

Advantages:Avoids the circularization step of iCLIP, which can be an unreliable reaction at times. This strategy also avoids the usage of radioactive markers.High-throughput mapping of protein-RNA binding sites.Ligation efficiency is better than the other methods.

Disadvantages:Extensive protocol and UV crosslinking may be poor.A single-stranded DNA adaptor must be ligated to single-stranded cDNA.

## 2. Yeast Three-Hybrid System

By using simple phenotypic assays, the three-hybrid system detects RNA–protein interactions in yeast [48]. This system monitors RNA–protein interactions by recording reporter gene output, permitting the analysis of an interaction independent of biological functions. Like other genetic strategies, the three-hybrid system yields a clone of the RNA or protein of interest during the screen, establishing its identity early in the process [29,48,49,50]. 

The general strategy of the three-hybrid system is shown in Figure 3. The DNA-binding sites are located upstream of a reporter gene, which is integrated into the yeast genome. The first hybrid protein contains a DNA binding domain that is linked to an RNA-binding domain. The RNA-binding domain interacts with its RNA-binding site through a bi-functional hybrid RNA molecule. The other part of the hybrid RNA interacts with a second hybrid protein containing another RNA-binding domain linked to a transcription activation domain. Once this tripartite complex is formed at the promoter, the reporter gene is activated. The phenotype or simple biochemical assays can detect the reporter gene expression. This assay’s commonly used components are a DNA-binding site with a 17-nucleotide recognition site for the *Escherichia coli* LexA protein. HIS3 and lacZ are the most widely used reporters in this assay. Control experiments are used to show that each component of the system is required for the assayed interaction. Reporter gene expression is assayed either by β-galactosidase activity or by determining the level of resistance to 3-aminotriazole 3-AT, which monitors His3 activity.

Advantages:This method can be used successfully for RNAs of different sizes 21–1600 nucleotides [51].There is no strict rule for the bait (internal or 3’) position in the hybrid RNA.The three-hybrid system allows for differential identification of the RNP complexes. This strategy helps to select proteins that bind to RNA at different steps in a functional pathway.

Disadvantages:RNAs with post-transcriptional modifications, necessary for interaction with their protein binding partners, cannot be studied using this method as the hybrid RNAs used here may be partially modified or unmodified.Many proteins require cofactors to bind to their RNA partners, and these cofactors might not be available in the nucleus because they are located in other cellular compartments.Some of the proteins and their target RNAs interact in a transient manner and with low affinity, thus preventing complexes’ detection with the three-hybrid strategy.Regarding the three-hybrid strategy, the RNA-protein interaction occurs in vivo, where many cellular parameters can influence it. These influences can lead to false-positive results, which additional time-consuming screening tests must eliminate.This strategy requires an effective fusion of the proteins and RNA, which renders a potential risk in altering the structures of the macromolecules, especially the RNA structure in the fusion.

## 3. Cell Mixing Experiments to Identify RNA–Protein Interactions In Vivo

The approaches to characterize RNA complexes associated with specific RNA-binding proteins depend on immuno- or affinity purification procedures and subsequent identification of the co-purifying RNA or protein molecules. Understanding these experiments is usually based on the assumption that the in vivo assembled RNA-protein complexes’ integrity is preserved during the investigation. However, the validity of this assumption has not been tested experimentally. The cell mixing strategy shows that an RNA-binding protein expressed in one cell population can associate with a target RNA expressed in a different cell population after cell lysis [52,53,54]. This finding indicates that co-immuno-precipitation may not always reiterate the in vivo state of RNP complexes.

Six different cultures, A to F, are used for this assay [37]. Culture A contains both the tagged protein and ligand RNA expression constructs. Culture B has corresponding empty vectors used in culture A for protein and RNA expression. The cells from culture A and culture B are mixed and lysed, and the tagged proteins are then immuno-precipitated. Culture C contains the ligand RNA expression construct and the tagged protein vector, whereas culture D contains the ligand RNA vector and tagged protein expression construct. After mixing culture C and D, lysis, and IP, the RNA is not expected to co-IP if the interaction is formed exclusively in vivo. However, if the RNA and protein factors interact after lysis, the RNA will co-IP from these extracts. Like other IP experiments, the last set of mixtures is performed, and neither group of cultures expresses the tagged construct culture E and F, thus allowing examination of the nonspecific interactions. If the mix of cultures C and D is negative, this last mixing step may be omitted. Therefore, if the co-IP efficiency from the mixture of cultures A and B is significantly increased compared with the mix of cultures C and D, it can be inferred that the tagged protein and the ligand RNA interact in vivo.

Advantages:The procedure is straightforward, without any crosslinking or stripping of the RNA from proteins. And cells are lysed by using relatively milder cell lysis procedures, which helps decrease RNA degradation and does not interfere with other downstream analyses.

Disadvantages:The major disadvantage of the cell mixing protocol is that it can rarely be applied to cellular RNA–protein interactions.

In contrast to the UV crosslinking discussed in method, one cannot determine whether an observed interaction between an RNA and protein is direct or indirect in the cell mixing assay.

## 4. Trimolecular Fluorescence Complementation TriFC to Detect RNA–Protein Interactions in Live Cells

Trimolecular fluorescence complementation TriFC is used to detect RNA–protein interactions in live cells [55]. In this approach, the MS2 coat protein and the protein of interest are attached to two complementary portions of the Venus fluorescent protein. The binding of the two fusion proteins to a substrate RNA containing the specific sequence and the MS2-binding site allows the two parts of Venus to come into proximity and emit fluorescence.

In this method, a portion of the Venus fluorescent protein [56] is attached to an mRNA of interest through the MS2 coat protein-RNA operator interaction, as in Figure 4A [55]. The complementary portion of the Venus protein is fused to an RNA-binding protein of interest. Suppose the RNA-binding protein is able to interact with the m/RNA sequence of interest. In that case, it will bring the two portions of the Venus protein into proximity and form a fluorescent complex Figure 4B, thus allowing the interaction site within the cell to be identified. The use of Venus, rather than yellow fluorescent protein, has advantages because it folds faster and has brighter fluorescence [56].

Advantages:This method enables the simple detection and localization of the interacting RNA and protein components in living cells where RNAs and proteins are expressed in their native locations, making this method both physically and biologically relevant.

Disadvantages:Similar to most other in vivo systems, this method can also be nonspecific at times. Because the complexes can form heterogeneous in natural environments, the endogenous proteins or RNAs can bring the TriFC components into proximity. In turn, this proximity enables the fluorescent signal to be detected in cases where the TriFC components do not directly interact but instead associate via a bridging RNA or protein. This problem can be resolved by analyzing purified components in vitro or by RNAi knockdown of the candidate bridging molecules.Although this method detects the RNA–protein interactions in situ, its other drawback is the irreversible crosslinking between the studied RNA-binding protein, MS2–protein, and likely the target RNA.BiFC protein’s maturation takes a long time after the formation of the protein–protein complex [57]. Therefore, sequential and spatial regulation in response to physiological events may be difficult to observe through this method.

## 5. Fluorescence Resonance Energy Transfer (FRET) and In Vivo Detection of RNA–Protein Interactions

Fluorescence resonance energy transfer (FRET) is a photophysical phenomenon in which energy is transported between two suitable fluorophores, a donor and an acceptor, appropriately oriented and located within a specific range usually, <10 nm. In cell biology, fluorophores are often used as tags on proteins, and their proximity is achieved through an interaction with the proteins. However, in FRET, the tagged proteins do not need to interact directly, and it is sufficient when FRET partners are part of the same complex. For example, FRET has successfully been used to measure protein–protein interactions between splicing factors in the spliceosome [58,59].

FRET was first used by Lorenz [60] to detect RNA–protein interactions inside the cell. However, the assay could not precisely determine whether the studied protein interacts with a specific substrate RNA. Later in that year, Huranova and co-workers [61] used a similar but modified technique to detect specific RNA–protein interactions inside the cell, called RNA-binding-mediated FRET RB-FRET.

This method has overcome some of the disadvantages of the TriFC assay. This experiment’s strategy is shown in Figure 4, in which the RNA-binding protein of interest contains an enhanced cyan fluorescent protein ECFP and is used as the FRET donor. In contrast, the MS2 coat protein is tagged with the enhanced yellow fluorescent protein EYFP and is used as the FRET acceptor. A target RNA containing the binding site that the RBP recognizes in the proximity of the high-affinity MS2 RNA operator is generated by a third construct [61]. FRET will result from simultaneous binding of the RBP-ECFP and MS2-EYFP fusion proteins to the target RNA. 

FRET signals between the ECFP-protein of interest and MS2-EYFP pairs are measured by the acceptor photobleaching method, which relies on detecting de-quenched donor fluorophores in the presence of acceptors [62] (Figure 5). Furthermore, in control experiments with non-RNA-RBPs, the absence of FRET signal should rule out any nonspecific interaction.

Advantages:

This strategy is specifically useful for studying RNA-binding proteins with known RNA target sequences.

As the studied RNAs and proteins can be expressed within a living cell, both the physical and biological properties of these complexes can be authentically examined, limiting in vitro assays.

## 6. Peptide Nucleic Acid (PNA)-Assisted Identification of RNA Binding Protein (PAIR) Technology to Identify Ribonucleoprotein–RNA Interactions In Vivo

The PAIR approach helps identify RBPs that interact with a specific target m/RNA sequence in vivo [63]. This approach is accomplished through a peptide nucleic acid (PNA) that is coupled to a cell membrane-penetrating peptide (CPP) and a photoactive compound. PNAs are nucleic acid analogs in which the sugar-phosphate backbone is substituted with a polyamide skeleton without varying the position of the nucleobases [64]. PNA binding to the RNA target in particular, and the rapidly formed PNA-RNA hybrids are more stable than the corresponding nucleic acid hybrids [64]. In combination with their resistance to proteases and nucleases [54], these characteristics have permitted PNAs to be used in various molecular biological and therapeutic contexts [65].

This strategy was first used by Zielinski and co-workers [63], who used transportan 10 TP10, a truncated analog that lacks GTPase activity of the full-length transportan protein. The authors used the *ankylosis ank* RNA, a pan-neuronal dendritically localized RNA, to develop PAIR. As shown in Figure 5, the *p*-benzoylphenylalanine (BPA)-PNA dissociates from CPP after entering into the cell (Figure 6). PNA hybridizes to its complementary sequence and positions the photo-sensitizing BPA in proximity to the target RNA UV irradiation crosslinks the adjacent substances, i.e., RBPs, with the PNA. After cell lysis and RNase treatment, the PNA-RBP complexes are isolated with magnetic beads coupled to a sense oligonucleotide. The isolated material is proteolyzed and analyzed by mass spectrometry. The same laboratory later used this method to isolate exon-specific RBPs from living cells [66].

Advantages:The PAIR technology offers a fast and reliable method to detect a subset of proteins that interact with RNAs in vivo.The ability to characterize the RBPs that bind to RNAs in vivo, with the capacity to detect the RNAs attached to any RBP in situ, should notice the dynamic changes in the RNA–RBP interactions are likely to control many cellular mechanisms.

Disadvantages:Crosslinking is irreversible.This method can be nonspecific for the targets.

## 7. Other Techniques Used to Assay RNA–Protein Interactions In Vivo

### 7.1. In Vivo Foot-Printing

The so-called foot-printing method is one of the most informative techniques to analyze proteins’ interaction with nucleic acids. It relies on protein-induced changes in the reactivity of nucleic acids toward a modifying agent. This technique has been instrumental in characterizing the large number of currently known DNA-binding proteins in eukaryotic cells. An analysis of the interaction of the proteins with RNA sequences using in vivo foot-printing involves the following two steps [67,68]: (i) in situ modification of the nucleic acids by the foot-printing reagent and (ii) visualizations of the footprints. This method is primarily used in yeast and mammals. RNaseT1 has been used to footprint live cells. The reaction conditions for in vivo foot-printing must be optimized to obtain a reliable impression. The modification of the target RNA should be as close as possible to single-hit, in order to avoid destabilization of the RNA-RNA or RNA-protein complexes. However, the extent of the digestion should be sufficient to provide a good signal-to-noise ratio, and there must be a sufficiently large number of modified molecules for statistical significance.

### 7.2. RNP Interaction Trap Assay RITA

The RNP-interaction trap assay is modified from the yeast three-hybrid assay, and it allows in vivo re-formation of tripartite protein-RNA-protein ribonucleoproteins RNPs. This technique has been used explicitly in the yeast system [69], in which the RNA acting as a bridge between the two hybrid proteins activates the reporter genes lacZ and HIS3 are commonly used. Thus, allowing yeast to gain the capacity to grow on a medium lacking histidine and to turn blue when 5-Bromo-4-chloro-3-indolyl-β-D-galactoside X-gal is added to the medium. Bouffard et al. have used rRo60 and rLa proteins co-expressed with recombinant hY RNAs in yeast. Reporter genes are activated, thus indicating that a tripartite RNP is reconstituted, and the rhY RNA acts as a bridge linking the rRo60 and rLa proteins [69].

### 7.3. Immunoprecipitation-RNA: rPCR for the In Vivo Isolation of RNP Complexes

In the field of RNA-protein complexes, IP techniques are developed to identify and characterize the components of small RNPs in mammalian cells [70]. However, this technique is not very sensitive in identifying the mRNA components of RNP complexes that are expressed at low levels inside the cell. The IP-RNA: random PCR rPCR method has been developed to detect and characterize RNP complexes’ cellular m/RNA components and overcome this limitation. This technique is based on the IP method introduced by Lerner and Steitz [71] and is combined with the integration of the R.N.A.: rPCR technique reported by Froussard [72] and Edward Chu and colleagues [73]. The strategy of coupling the rPCR amplification method to an IP assay has several essential benefits over IP. The goal of this procedure is to isolate cellular RNAs in RNPs. Thus, the reagents must be RNase-free to minimize the risk of RNA degradation during the process. 

### 7.4. SHAPE-MaP Strategy Based Detection of RNA-Protein Interaction in Living Cells

SHAPE-MaP (selective 2-hydroxyl acylation analyzed by primer extension and mutational profiling) uses a highly validated SHAPE RNA structure probing method followed by selective sequencing to achieve high-throughput examination of RNA flexibility at single-nucleotide resolution [74,75]. In SHAPE experiments, 2″-hydroxyl-selective reagents react to form covalent 2″-*O*-adducts at conformationally flexible RNA nucleotides, both in vitro and in vivo [75,76]. In brief, total RNA, in the cell or isolated under native conditions, is treated with a SHAPE reagent, 1-methyl-7-nitroisatoic anhydride (1M7) [75], where in vitro and in vivo RNA–protein interactions often exhibit different SHAPE profiles (low, moderate, and high interaction). Once treated, the difference in SHAPE reactivities (ΔSHAPE) is calculated between in vivo and in vitro conditions for each nucleotide to ultimately calculate the Z-factor (detailed explanation of equation in [75]. Positive ΔSHAPE values indicate protection from modification in the cellular environment, whereas negative ΔSHAPE signifies enhanced reactivity in cells. Nucleotides showing *Z* > 0 (more than 1.96 standard deviations when comparing the two conditions) are considered to undergo significant changes in SHAPE reactivity.

## 8. Concluding Remarks

The studies described in this report have successfully proven efficient for analyzing RNAs of different sizes and their interactions with target proteins. Each assay has its advantages and disadvantages (Table 1). To date, in vivo crosslinking and subsequent immunoprecipitation strategies are the most commonly used techniques. However, depending on the nature of the cells, the selection of methods follows. Interestingly, there are far fewer in vivo than in vitro techniques. We tried to focus on in vivo techniques, which are generally used to detect RNA–protein interactions and identify the components of RNPs. Some of the methods are specific and do not have a broad range of use in prokaryotes and eukaryotes. However, these protocols allow a great deal of knowledge and provide an excellent starting point for investigating many types of RNA–protein interactions.

## Figures and Tables

**Figure 1 mps-04-00022-f001:**
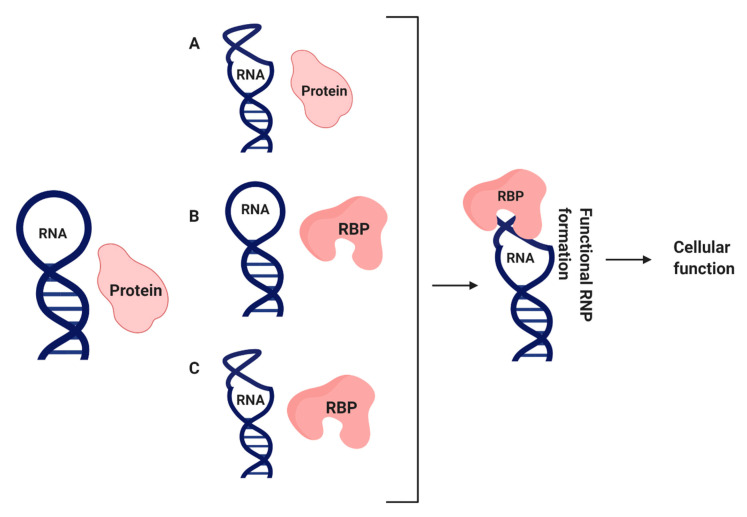
Observed conformational changes during RNA–protein interactions. Protein-induced RNA folding (**A**), RNA-induced protein folding (**B**), and co-induced folding (**C**) [3].

**Figure 2 mps-04-00022-f002:**
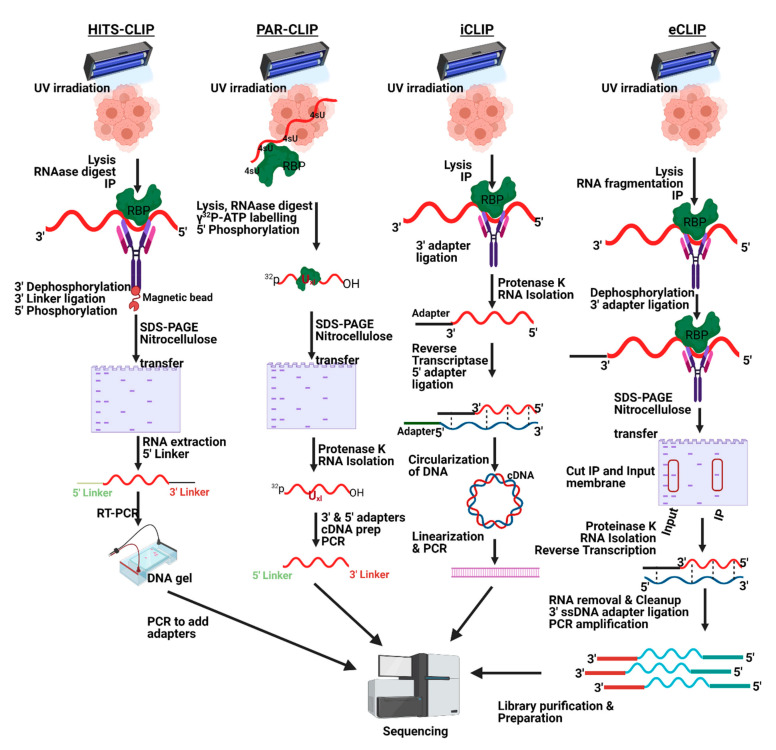
Comparison between the crosslinking and immunoprecipitation (CLIP) protocols.

**Figure 3 mps-04-00022-f003:**
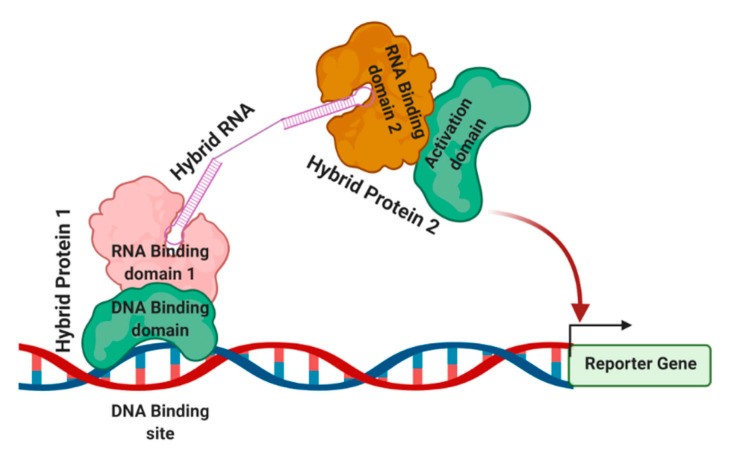
Three-hybrid system to detect and analyze RNA–protein interactions. A three-hybrid system detects RNA–protein interactions. The hybrid RNA interacts with two separate proteins with RNA binding domains that independently interact with proteins containing DNA binding domain and an activation domain, respectively. Once this tripartite complex is formed successfully at the promoter, the reporter gene is activated, serving as a detection method [29,48].

**Figure 4 mps-04-00022-f004:**
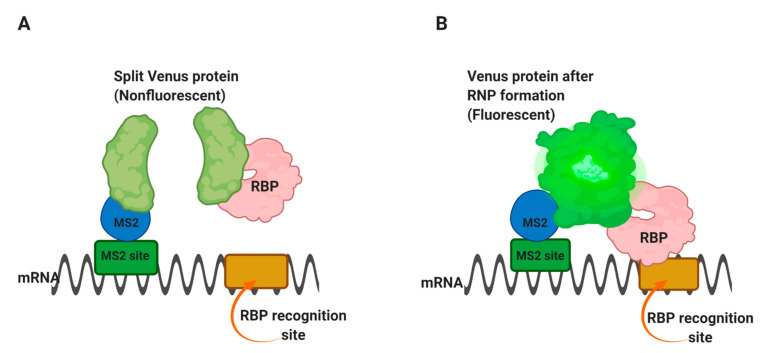
TriFC method in living cells. The TriFC helps detect RNA-protein interaction in living cells. (**A**) Two complementary portions of the Venus fluorescent protein are attached to a reporter mRNA by the MS2 coat protein and an RNA-binding protein, respectively. (**B**) If the RNA-binding protein finds a preferred sequence within the reporter mRNA and binds there, the two portions of Venus protein will be brought into proximity to form a fluorescent product [55].

**Figure 5 mps-04-00022-f005:**
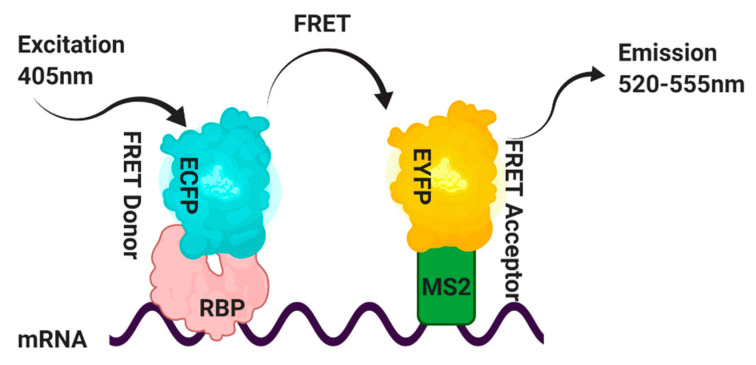
Schematic representation of the strategy utilized to visualize RNA–protein interactions in cells. Fluorescence resonance energy transfer (FRET) signals between the RNA-binding protein (RBP) (protein of interest) bound to the ECFP and MS2-EYFP pairs are detected by the acceptor photobleaching method. This method detects the de-quenched donor fluorophores in the presence of acceptors [62].

**Figure 6 mps-04-00022-f006:**
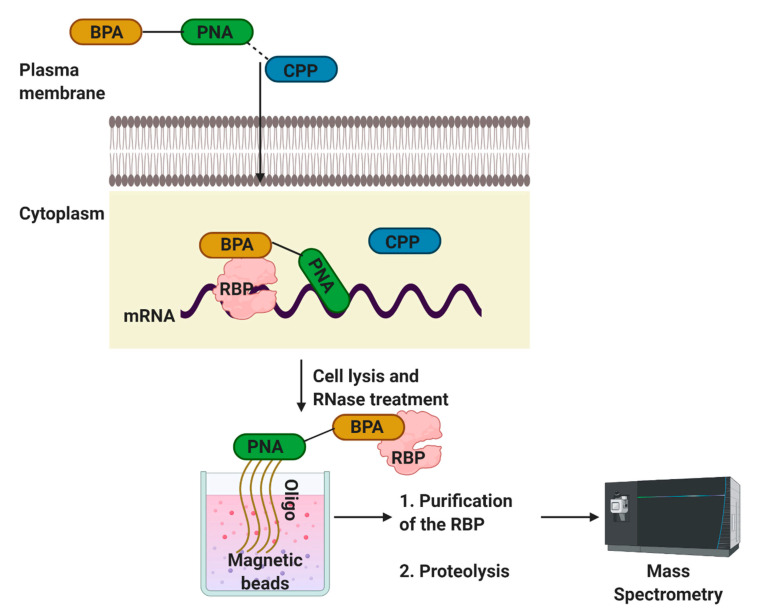
Schematic of the PAIR technology. A method using peptide-nucleic-acid-assisted identification of RBPs [63,66]. Peptide nucleic acid (PNA), cell membrane-penetrating peptide (CPP), *p*-benzoylphenylalanine (BPA).

**Table 1 mps-04-00022-t001:** Features of selected RNA–protein interaction methods.

Approach	Applications	Advantages	Disadvantages	Final Approach	Refs.
RNA Immunoprecipitation and RNA pull-down	In vitro and In vivo	No need for chemical or UV cross-linkers The binding of biotinylated RNA with streptavidin beads is very efficient	RNA and protein must have a strong affinity Efficiency is very low compare to other crosslinking agents In vitro, the approach is favored towards the enriched protein of interest in cell extracts	RNA-seq RT-PCR MS	[11,12,13,14,15,16,17,18,19,20,21,22,23,24,25,26,27,28,29,30]
Individual-nucleotide resolution Cross-Linking and Immuno Precipitation (iCLIP)	In vivo	UV and chemical cross-linking create strong interaction between RNA and protein Formaldehyde cross-linking is reversible, which is useful for RNA estimations UV makes a covalent bonding between RNA and protein	Time-consuming protocol The efficiency of binding between RNA and protein is low compare to chemical crosslinking Irreversible crosslinking followed by adducts formation disrupts the RT-PCR reaction at the reverse transcriptase step.	RNA-seq RT-PCR	[9]
eCLIP	In vivo	Mapping of protein-RNA binding sites are high throughput in nature Barcoded adapters reduce the noise of the PCR reaction	Nonspecific binding between antibodies to proteins forms precipitates makes it harder to separate the complexes Failure to detect all the RNA-binding protein domains UV crosslinking may be poor	RNA-seq RT-PCR	[47]
HITS-CLIP/PAR-CLIP	In vivo	Short protocol and very efficient ligation reactions produce very high reliability Low background noise and higher resolution of the binding site due to RNase digestion High accuracy in detecting RNA-protein associations	UV crosslinking may be inefficient Reverse transcriptase may have issues with cross-linked RNA On-bead ligation is less efficient for RNAs Cellular toxicity of 4-SU analogs are noted	RNA-seq RT-PCR	[2,46]
Yeast 3-hybrid	In vivo	Up to 1600, BP RNAs can be detected Determines the protein that binds to RNA	Failure to detect the RNA modifications Ribonucleoprotein complex may not work; only a single protein can be used Low-affinity proteins fail to detect RNA interactions	Reporter assays using lacZ or His3	[29,48,49,50,51]
Trimolecular fluorescence complementation	In vivo	Enable to detect the localization of RNA and protein of interest in live cells	Nonspecific interaction is a big issue Irreversible cross-linking is a problem The temporal and spatial arrangement of RNA-protein complex detections are hard	In situ hybridization Florescence microscopy	[55,56,57]
Fluorescence resonance energy transfer FRET	In vivo	Known RNA motifs or sequences can be detected through the known protein of interest	Very old technique and mainly used for protein–protein interactions Adaptation to detect RNA-protein interaction studies is very minimal	Fluorescence reporter assays	[58,59,60,61,62]

## Data Availability

Not applicable.

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
