# Peer review of "Compendium of Methods to Uncover RNA-Protein Interactions In Vivo"

_mps, 2021, doi:10.3390/mps4010022_

Round 1

Reviewer 1 Report

  • The figure 1 is confused and the legend does not represent the picture.
    I suggest adding a table informing details of each method, including if the interaction occurs in in vivo conditions, the pros, and cons, with or without cross-link, type of final data (example: Mass-spec, sequencing), or if works confirming previosly known interactions, and the original papers that describes each strategy.
    Line 485 - error in text formatting
    I miss the supplementary material 

Author Response

Response to Reviewer 1:

The figure 1 is confused and the legend does not represent the picture.

It was a mistake. We have now corrected to figure that matches with the figure 1 legend.

I suggest adding a table informing details of each method, including if the interaction occurs in in vivo conditions, the pros, and cons, with or without cross-link, type of final data (example: Mass-spec, sequencing), or if works confirming previously known interactions, and the original papers that describes each strategy.

We have now added a figure in the concluding remarks section that touches base with all the methods discussed here with appropriate references.

Line 485 - error in text formatting

In our original copy, line 485 falls within the bibliography section, which is automatically loaded by Endnote.

I miss the supplementary material 

There was no supplementary material.

Reviewer 2 Report

In the review the authors summarize the current state of methods to study protein-RNA interactions. In general, the review gives a nice overview and lists advantages and disadvantages of the methods discussed. However, the authors should consider the balance of detail discussed as I could not figure out at times if the manuscript is aiming more for a protocol style or a discussion of potential results. Besides this, I have a few comments the authors should think about and/or amend before publication:   - 2nd sentence: although it is true the RNA is more versatile than DNA, this sentence sounds odd as DNA can clearly be bound by a multitude of different proteins   - paragraph 2: proteomics is not a sequencing method   - the authors state that in vivo methods are not available in large numbers, but start discussing RNA´protein interactions using CLIP, which maps RNA-protein interactions in vivo   - it is weird that exogenous expression of tagged proteins is discussed so extensively. A large fraction of published CLIP data stems from IPs with antibodies raised against the protein itself - one might even argue that over-expression systems are not really perfect to study protein-RNA interaction as they can skew interaction profiles per se. This should be clarified.    - one thing that has not been discussed is antibody specificity that most techniques rely on   - iCLIP is characterized as a the method of choice, but there are points that should be listed as disadvantages:   -iCLIP needs radioactivity - I would argue that circularisation can be very inefficient compared to adapter ligation (van Nostrand et al., 2016), which is one of the reasons why eCLIP is used in a lot of labs nowadays (which is not discussed at all)   - the section about the yeast two hybrid is very good, but one point that should be mentioned is the fact that fusions of proteins/RNA have the potential risk to alter the structure (particularly for RNA) in the fusion  

Author Response

Response to Reviewer 2:

In the review the authors summarize the current state of methods to study protein-RNA interactions. In general, the review gives a nice overview and lists advantages and disadvantages of the methods discussed. However, the authors should consider the balance of detail discussed as I could not figure out at times if the manuscript is aiming more for a protocol style or a discussion of potential results. Besides this, I have a few comments the authors should think about and/or amend before publication:  

- 2nd sentence: although it is true the RNA is more versatile than DNA, this sentence sounds odd as DNA can clearly be bound by a multitude of different proteins  

We have now changed the sentence to, “Almost all RNAs, irrespective of their structure and function, interact with one or more protein partners to function properly (1, 2).”

- paragraph 2: proteomics is not a sequencing method  

We have removed proteomics from that sentence.

- the authors state that in vivo methods are not available in large numbers, but start discussing RNA´protein interactions using CLIP, which maps RNA-protein interactions in vivo   - it is weird that exogenous expression of tagged proteins is discussed so extensively. A large fraction of published CLIP data stems from IPs with antibodies raised against the protein itself - one might even argue that over-expression systems are not really perfect to study protein-RNA interaction as they can skew interaction profiles per se. This should be clarified.    

We have now added this section to the review. “Besides crosslinking efficiency, antibody specificity against the interacting protein is a critical factor in these assays as well. This can be overcome by the addition of epitope tags to the interacting proteins; however, it may restrict the proper folding of the protein inside the cells or affect the protein’s ability to interact with specific RNAs.”

- one thing that has not been discussed is antibody specificity that most techniques rely on   - iCLIP is characterized as a the method of choice, but there are points that should be listed as disadvantages:   -iCLIP needs radioactivity - I would argue that circularisation can be very inefficient compared to adapter ligation (van Nostrand et al., 2016), which is one of the reasons why eCLIP is used in a lot of labs nowadays (which is not discussed at all)  

We have added the disadvantage to the iCLIP section. “And the circularization step of iCLIP can be an unreliable reaction at times.”

We have also added a new section of eCLIP in the review.

- the section about the yeast two hybrid is very good, but one point that should be mentioned is the fact that fusions of proteins/RNA have the potential risk to alter the structure (particularly for RNA) in the fusion  

We have added this disadvantage of the yeast three-hybrid strategy in the review. “This strategy requires an effective fusion of the proteins and RNA, which renders a potential risk in altering the structures of the macromolecules, especially the RNA structure in the fusion.”

Reviewer 3 Report

This review titled: “Compendium of methods to uncover RNA-protein interactions in vivo” by Majumder and Palanisamy discusses the past and currently established methodologies that are used to study RNA-protein interactions in vivo.

This reviewer has the following major comments:

1) Delta SHAPE-MaP methodology has been developed to provide comprehensive insight into the RNA-protein interaction landscape inside the living cells. The discussion of this method as well as proper references should be included. See below:

Matthew J Smola  , J Mauro Calabrese , Kevin M Weeks. Detection of RNA-Protein Interactions in Living Cells with SHAPE. Biochemistry 2015;54(46):6867-75.  doi: 10.1021/acs.biochem.5b00977.

2) In chapter 2.1. the authors discussed methods to study RNP complexes and RNA. Some of the indicated methodologies involved adding Tags to RNA, e.g., Strepto or Sephadex tags. The authors should indicate potential disadvantages of that approach, i.e., distortions of RNA native fold, that must be controlled either by running RNA probing experiments or at the least by performing in silico RNA secondary structure predictions

3) In chapter 2.1., the authors should include the other means of tagging and capturing RNA molecules, i.e., use of antisense oligonucleotides (ASOs), aptamers

4) Chapter 2.2.1. Photo cross-linking (UV). The authors claimed should include reference that studied the efficiency of UV crosslinking. See:

E.C. Urdaneta, C.H. Vieira-Vieira, T. Hick, H.-H. Wessels, D. Figini, R. Moschall, J. Medenbach, U. Ohler, S. Granneman, M. Selbach, B.M. Beckmann. Purification of cross-linked RNA-protein complexes by phenol-toluol extraction. Nat. Commun., 10 (1) (2019), p. 990, 10.1038/s41467-019-08942-3.

Castello, B. Fischer, K. Eichelbaum, R. Horos, B.M. Beckmann, C. Strein, N.E. Davey, D.T. Humphreys, T. Preiss, L.M. Steinmetz, J. Krijgsveld, M.W. Hentze. Insights into RNA biology from an atlas of mammalian mRNA-binding proteins. Cell, 149 (6) (2012), pp. 1393-1406, 10.1016/j.cell.2012.04.031

Also, the biases for RNA nucleotides should be clearly stated. In general, in nucleic acids, pyrimidines are much more efficiently cross-linked in general than purines and with RNA being more reactive than DNA (poly rU > poly rC > poly dT > poly rA) when comparing addition to cysteine. See: K.C. Smith, D.H. Meun. Kinetics of the photochemical addition of [35S] cysteine to polynucleotides and nucleic acids Biochemistry, 7 (3) (1968), pp. 1033-1037

5) Chapter 2.3.1, The authors stated: “There are no amino acid preferences for forming the covalent crosslinks, making this assay independent of a particular conformation of the RNA-protein interface, unlike UV crosslinkin”. This is incorrect, with formaldehyde

cross-linking, strongly nucleophilic lysine residues are preferentially cross-linked. See: Hoffman EA, Frey BL, Smith LM & Auble DT Formaldehyde crosslinking: a tool for the study of chromatin complexes. J. Biol. Chem. 290, 26404–26411 (2015).

6) For chapters 2 discussing iCLIP, PAR-CLIP, HITS-CLIP, a general scheme comparing the steps of these methods would be very helpful

Minor suggestions:

Grammatic errors convolute the clarity of text, please address that.

Fig.1: font size is too small to read, and differs from other illustrations

Unify high-throughput vs high throughput

Expand all used abbreviations the first time they are cited in the text, e.g., miCLIP

Fig. 5 Caption is inadequate, none of the used abbreviations that appear in the illustrations are expanded, e.g., BPA, PNA, CPP

Chapter 8.3., the reference [62] still includes last names of cited authors, also wrong font size

Author Response

Response to Reviewer 3:

This review titled: “Compendium of methods to uncover RNA-protein interactions in vivo” by Majumder and Palanisamy discusses the past and currently established methodologies that are used to study RNA-protein interactions in vivo.

This reviewer has the following major comments:

1) Delta SHAPE-MaP methodology has been developed to provide comprehensive insight into the RNA-protein interaction landscape inside the living cells. The discussion of this method as well as proper references should be included. See below:

Matthew J Smola  , J Mauro Calabrese , Kevin M Weeks. Detection of RNA-Protein Interactions in Living Cells with SHAPE. Biochemistry 2015;54(46):6867-75.  doi: 10.1021/acs.biochem.5b00977.

We thank the reviewer for this critique. We have now added this methodology to the review. Please see section 7.4 with appropriate references.

2) In chapter 2.1. the authors discussed methods to study RNP complexes and RNA. Some of the indicated methodologies involved adding Tags to RNA, e.g., Strepto or Sephadex tags. The authors should indicate potential disadvantages of that approach, i.e., distortions of RNA native fold, that must be controlled either by running RNA probing experiments or at the least by performing in silico RNA secondary structure predictions

We have added these sentences in section 1.1. “RNA affinity tags (aptamers) like Strepto, Streptavidin S1, and Sephadex D8 are commonly and successfully used to isolate the in vivo RNA-protein complexes. However, due to the addition of any tags to the macromolecules, distortions of the RNA native fold may occur. In silico RNA secondary structure must be verified after adding a tag to the RNA by running RNA probing experiments (22, 23).”….” Sephadex G-200 is inexpensive, and the concentration of ligand on the beads is almost infinite, which makes this purification method very useful when large starting quantities are employed. However, unlike streptavidin-agarose, the affinity of RNA for the ligand is not high, which increases the chance of loss of the bound complex to the G-200 resin after extensive washing.”

3) In chapter 2.1., the authors should include the other means of tagging and capturing RNA molecules, i.e., use of antisense oligonucleotides (ASOs), aptamers

We have added a paragraph including ASOs and TRIP procedures. Please see lines from 105-119.

4) Chapter 2.2.1. Photo cross-linking (UV). The authors claimed should include reference that studied the efficiency of UV crosslinking. See:

E.C. Urdaneta, C.H. Vieira-Vieira, T. Hick, H.-H. Wessels, D. Figini, R. Moschall, J. Medenbach, U. Ohler, S. Granneman, M. Selbach, B.M. Beckmann. Purification of cross-linked RNA-protein complexes by phenol-toluol extraction. Nat. Commun., 10 (1) (2019), p. 990, 10.1038/s41467-019-08942-3.

Castello, B. Fischer, K. Eichelbaum, R. Horos, B.M. Beckmann, C. Strein, N.E. Davey, D.T. Humphreys, T. Preiss, L.M. Steinmetz, J. Krijgsveld, M.W. Hentze. Insights into RNA biology from an atlas of mammalian mRNA-binding proteins. Cell, 149 (6) (2012), pp. 1393-1406, 10.1016/j.cell.2012.04.031

Thank you. All the references have been added to the review. Ref. 35 and 39.

Also, the biases for RNA nucleotides should be clearly stated. In general, in nucleic acids, pyrimidines are much more efficiently cross-linked in general than purines and with RNA being more reactive than DNA (poly rU > poly rC > poly dT > poly rA) when comparing addition to cysteine. See: K.C. Smith, D.H. Meun. Kinetics of the photochemical addition of [35S] cysteine to polynucleotides and nucleic acids Biochemistry, 7 (3) (1968), pp. 1033-1037

5) Chapter 2.3.1, The authors stated: “There are no amino acid preferences for forming the covalent crosslinks, making this assay independent of a particular conformation of the RNA-protein interface, unlike UV crosslinkin”. This is incorrect, with formaldehyde

cross-linking, strongly nucleophilic lysine residues are preferentially cross-linked. See: Hoffman EA, Frey BL, Smith LM & Auble DT Formaldehyde crosslinking: a tool for the study of chromatin complexes. J. Biol. Chem. 290, 26404–26411 (2015).

We have added a section covering both the comments above, “In proteins, the amino acids cysteine, tyrosine, phenylalanine, arginine, lysine, and tryptophan are the most readily crosslink to poly-U. Additionally, it is often noticed that pyrimidines have higher efficiency to be crosslinked than purines. Thus, RNA, with uridine residues compared to thymidines in DNA, shows greater reactivity when comparing addition to cysteine (33, 34).”

6) For chapters 2 discussing iCLIP, PAR-CLIP, HITS-CLIP, a general scheme comparing the steps of these methods would be very helpful

We have added a figure (figure 2) showing a flowchart comparison of the CLIP protocols with newly added eCLIP.

Minor suggestions:

Grammatic errors convolute the clarity of text, please address that. We have corrected any significant grammatical issues that we could find.

Fig.1: font size is too small to read, and differs from other illustrations. At our end, we could not find a difference in font size between figure 1 and the rest of the figures.

Unify high-throughput vs high throughput. We have corrected that.

Expand all used abbreviations the first time they are cited in the text, e.g., miCLIP. Corrected

Fig. 5 Caption is inadequate, none of the used abbreviations that appear in the illustrations are expanded, e.g., BPA, PNA, CPP. We have extended all the abbreviated names.

Chapter 8.3., the reference [62] still includes last names of cited authors, also wrong font size. Corrected.

Round 2

Reviewer 3 Report

The authors addressed all the issues pinpointed by the reviewer, therefore, the manuscript should be accepted in current form.